# A New Perspective for Improving the Human Resource Development of Primary Medical and Health Care Institutions: A Structural Equation Model Study

**DOI:** 10.3390/ijerph18052560

**Published:** 2021-03-04

**Authors:** Huanhuan Jia, Peng Cao, Jianxing Yu, Jingru Zhang, Hairui Jiang, Qize Zhao, Xihe Yu

**Affiliations:** 1School of Public Health, Jilin University, Changchun 130000, China; hhjia20@mails.jlu.edu.cn (H.J.); cppengcao@163.com (P.C.); yjxjlu@163.com (J.Y.); 18834184124@163.com (J.Z.); jianghairuii@163.com (H.J.); 2Jilin Province Healthcare Security Administration Management Center, Changchun 130000, China; qzzhao17@mails.jlu.edu.cn

**Keywords:** health workforce, primary medical and health care institution, implicit theory, lexical approach

## Abstract

In some countries, including China, primary health care is rarely utilized because of medical personnel shortages at primary medical and health care institutions (PMHCIs). Several studies suggest that the most effective solution is to guide qualified doctors and medical graduates to work in PMHCIs, but the studies and measures have been formulated only from the perspective of the government and PMHCIs; few have considered the subjective willingness of medical personnel. Therefore, it is necessary to explore the measures to develop human resources of PMHCIs from the guiding object. This research was divided into two parts based on implicit theory and a lexical approach. The first part collected the factors affecting their choosing PMHCIs for employment, and the second part used exploratory factor analysis (EFA), confirmatory factor analysis (CFA), and structural equation modeling (SEM) to explore the dimensions and paths of the influencing factors. At last, seven factors were obtained from the EFA, and the SEM hypothesis fit the data well. Internal Organization Development, Patient Factor, Remuneration and Development, and Family Support had a significantly positive effect on the Sense of Gain of medical personnel seeking employment at PMHCIs, whereas both Job Responsibilities and Condition of the City Where the PMHCI Is Located had no significant effect. In addition, the indirect effects of Internal Organization Development and Condition of the City Where the PMHCI Is Located on the Sense of Gain were significant. The Patient Factor, Family Support, and Remuneration and Development significantly mediated the relationship between the internal and external environment of the institution and the Sense of Gain, whereas the mediating effect of Job Responsibilities was not significant. The improvement of family support, remuneration and development, and patient factors increase the willingness of medical personnel to seek employment at PMHCIs. In addition, the internal and external environments of a PMHCI play a vital role in guiding medical personnel to PMHCIs for employment. This research provides theoretical support for improving the development of human resources, guiding medical personnel to work in PMHCIs, and promoting the use of primary care services.

## 1. Introduction

Primary health care has attracted attention in many countries, and a series of measures have been taken to promote access to primary health care, such as India’s rural health centers [1], Nepal’s primary healthcare (PHC) system [2], and the Affordable Care Act (ACA) in the United States [3]. In China, primary health care is undertaken by primary medical and health care institutions (PMHCIs), primarily community health service centers and township hospitals, which provide general clinical care and basic public health services to residents [4,5]. Recognizing the double burden of the prevalence of chronic noncommunicable diseases [6] and increasing health costs [7] caused by population aging [8,9], behavioral changes [10], and rapid urbanization [11], the Chinese government began to reform the medical and health care system in 2009. In this reform, the improvement of the primary medical service system was one of the five key issues [5,12], and a hierarchical medical system was proposed to effectively balance various medical service resources and divert the general outpatient, rehabilitation, and nursing care originally undertaken by large and medium-sized medical institutions to PMHCIs [13,14], highlighting the role of these facilities. Since then, a considerable amount of money has been invested to improve the service capacity and quality of PMHCIs [15,16], and as a result, the facilities and environment of PMHCIs have been improved. However, the services provided by PMHCIs are rarely utilized [17,18], and the operation of PMHCIs is poor because of the lack of service uptake by patients [13,19]. Previous surveys [20,21] have shown that patients’ distrust of PMHCIs is one important reason for this, and the paucity and low technical level of medical personnel caused by their attrition is one of the important sources of this distrust. Therefore, increasing the number and technical capacity of medical personnel [22] in PMHCIs is an urgent measure to promote the use of primary health care.

The Chinese government has developed various types of training and practice systems for general practitioners to strengthen the human resource development of PMHCIs [23]. Previous studies have proposed that the government and PMHCIs should focus on addressing medical personnel’s work status and work-related demands [24]; balancing their work pressures [25]; improving their financial remuneration [26]; providing more opportunities for learning, training, and individual development [27]; increasing their work passion and enthusiasm; reducing their attrition; and improving their technical skill level. However, it will take a long time to train a sufficient number of general practitioners [13], and formulating incentives to reduce the loss of medical personnel cannot completely compensate for the shortage of medical personnel in PMHCIs. Therefore, we recommended that the most effective solution is to guide qualified doctors [28] and medical graduates [13] to seek employment at PMHCIs. In China, the candidates who take the medical licensing examination have received systematic clinical skills training, and most of them are medical graduates or unemployed interns who are seeking a job; it is this group that is most susceptible to guidance. In addition, we found that previous studies lacked in-depth research on specific guidance measures, and most of the measures were formulated from the perspective of the government and PMHCIs without comprehensive exploration and discussion of various factors such as family, the local economy, and culture from the perspective of medical personnel. We emphasize that medical personnel in PMHCIs are not only the main providers of primary health care but also direct participants in promoting the reform of China’s medical and health system; thus, the role of medical personnel should be changed from passive recipients of policies to participants in making policies. Therefore, this paper took candidates for the medical licensing examination as the research subjects and analyzed the factors that affected their choice to seek employment at PMHCIs and the relationships among them from the guided objects’ own perspective.

Implicit theory refers to people’s views on the conceptualization, structure, and development process of certain psychological characteristics that are formed in the context of daily life and work and exist in the individual’s mind in some form and can accurately and fully reflect people’s mental representations [29]. The role of implicit theory in information organization and interpretation has been increasingly accepted by cognitive psychologists and social psychologists [30] and has been extended to self-regulation [31], leadership effectiveness [32], emotional and mental health [33], and wisdom [34] in recent years. In addition, the lexical approach was originally used by psychologists to study personality traits, and it is based on the hypothesis that the individual differences that are of most significance in the daily transactions of persons with each other eventually become encoded in their language [35,36]. The lexical approach provides a research strategy aimed at identifying a set of relatively small, roughly independent axes along which people’s typical behavioral tendencies differ [35]. Given that the crucial role that organizational entities play in individuals’ lives and works, the words that people use to describe organizations when talking to others would also been encoded in the language. Similar to the concept of personality traits, organizational descriptors would be those used by people to distinguish one organization from another and differentiate people’s opinions about the same organization [37]. Therefore, the lexical approach can be used to collect the organization descriptors of a certain group of people on a particular organizational entity or a type of organizational entity, and then understand the group’s perception toward activities related to that organizational entity. The argument here is that in the general context of China’s medical and health reform, candidates are definitely exposed to information about PMHCIs in their lives and work, and this information is encoded in their minds and language, manifesting in candidates’ attitudes and views toward PMHCIs.

Therefore, implicit theory was applied as the starting point, and the lexical approach was used to explore the attitudes of candidates who were taking the medical licensing examination regarding seeking employment at PMHCIs. This research was divided into two parts according to the procedure of lexical studies [38,39]. The first step was a pre-investigation to collect influencing factors, and the second step was a formal investigation to explore the dimensions and relationships of the influencing factors.

## 2. Pre-Investigation

### 2.1. Participants and Procedures

Due to the brevity of the exam period, the pre-investigation and formal investigation could not be completed at the same time, so we chose an alternative population in which to conduct the pre-investigation. Given that students and interns who are about to graduate or are participating in standardized resident training at a hospital are either preparing for or have just finished the medical licensing examination and are searching for a job, their profile is consistent with that of the groups targeted by our research. Therefore, groups of students and interns from two hospitals who were involved in medical, teaching, and scientific research tasks were selected as the subjects of the pre-investigation.

A data collection service from a leading Chinese online survey site was adopted to administer the pre-investigation. First, we contacted the hospital training department that manages students and interns, requested that the online questionnaire be sent to the target group, and asked the respondents to complete the survey within a certain period of time. A total of 367 students and interns participated in the pre-investigation, but 15 people had signed an employment contract with a hospital, 5 were majoring in preventive medicine, and 3 filled in “None” for influencing factors. Ultimately, 344 complete and valid responses were obtained, and the effective response rate was 93.66%. In this sample, most of the respondents were women (74.71%), were majoring in clinical medicine (90.70%), and had a college or graduate degree (98.06%); the mean age was 25.76 (± 2.81). Our study procedures were approved by the Medical Ethics Committee of the School of Public Health, Jilin University (No. 20181102). According to ethical standards and practices, participants received a complete explanation of the research purpose. In addition, they were told that the information collected would be used only for research purposes and that they could withdraw from the study at any time.

### 2.2. Measures

A questionnaire containing open-ended questions was used, and the respondents were asked to answer the following question: “If a PMHCI was recruiting medical personnel, what factors would influence you to seek employment with a PMHCI? Please write down these factors in simple words or phrases.” The questionnaire also collected the sociodemographic characteristics of the students and interns, including gender, age, marital status, education level, and major. To ensure data quality and reduce social desirability bias, we adopted several relevant measures in the questionnaire design and data cleaning. For example, we set up screening questions to ensure that the answers came from eligible respondents, including “Are you a regular employee of the hospital?”, “Have you signed an employment contract with a hospital?”, and “What is your major?”. In addition, answering time, empty item review and IP confirmation (i.e., the same computer restriction and geographic restriction were used to prevent one respondent from filling out the questionnaire multiple times and people from other regions from filling out the questionnaire) were adopt to ensure the quality of the data collected.

### 2.3. Data Analysis

Ultimately, 1439 items were obtained from the pre-investigation. However, many words were repeated, and some concepts were expressed in complex phrases or sentences, so we cleaned and organized the lexical data without changing the meaning. The principles were as follows: (1) Remove modifiers such as adjectives and adverbs to extract key information; for example, “High wage level” and “Well-paid” were recoded as “Wage”, and “The primary medical institution is close to home” was recoded as “Distance from home”. (2) Combine synonymous words; for example, “High wage level” and “Well-paid” were combined into “Wage”, and “House issues” and “Accommodation environment” were combined into “Housing”. (3) Split combined concepts; for example, “What is the environment and economy of the institution’s location” was divided into “The environment of the city where the PMHCI is located” and “The economy of the city where the PMHCI is located”, and “Whether my wife and parents agree” was divided into “Spouse” and “Parents”. (4) Delete words obviously irrelevant to our research, such as the names of people, places, and institutions.

Finally, all factors were recoded into 103 items. The results showed that 51 of these items had a frequency of less than 4; that is, these factors were mentioned by less than 1% of the students and interns. Because of their lack of representativeness, these items were deleted from the formal investigation. Finally, without changing the meaning of the vocabulary, 52 factors were adjusted and normalized and then used for the preparation of the questionnaire.

## 3. Investigation

### 3.1. Participants and Procedures

The subjects of the investigation were all candidates who signed up for the medical licensing examination at a site in Changchun, Jilin Province. Our study procedures were approved by the Medical Ethics Committee of the School of Public Health, Jilin University (No. 20181102) too. The investigators were a postgraduate team from the School of Public Health at a university. We contacted the Changchun Medical Examination Center and asked it to cooperate with us in completing the research work. During the investigation, the investigator explained in advance the purpose and filling requirements of the questionnaire and informed the candidates that participation was voluntary. The questionnaires were then distributed to the candidates and retrieved immediately upon completion.

Finally, a total of 1883 questionnaires were distributed, all questionnaires were returned, and 105 questionnaires were incomplete or had the same answers for all items, which were deleted from the analysis (the completion rate was 94.42%). Ultimately, 1778 complete and valid responses were obtained. All paper questionnaires were manually entered into a computer database. In this sample, most of the respondents were women (60.74%), were majoring in clinical medicine (81.72%), had a college degree (80.82%), and were studying or working in tertiary hospitals (64.64%); the mean age was 29.08 (±5.57).

### 3.2. Measures

The 52 factors from the pre-investigation were used to create the main part of the questionnaire. The question was “If a PMHCI was recruiting medical personnel, please rate the importance of the following factors in their influence on your choice to seek employment at a PMHCI based on your actual situation”, and we used a 5-point Likert scale to measure the factors, with anchors ranging from 1 (very unimportant) to 5 (very important). Further, the sociodemographic characteristics of the respondents were collected by the questionnaire (gender, age, degree, major, and study or work institution).

### 3.3. Data Analysis

To explore the factors influencing medical personnel to seek employment at a PMHCI and their relationship, we needed to not only explore the underlying structure of the factors but also to conduct a hypothesis test on this structure. Therefore, in the analysis process, half of the sample was chosen randomly for exploratory factor analysis (EFA), which was used to extract common factors from the items and obtain concise and representative factors and then put forward a hypothesized model. Then, for the other half of the sample, confirmatory factor analysis (CFA) was adopted to test the reliability and validity of the dimensions, and finally, structural equation modeling (SEM) was applied to test the hypothesis.

Specifically, EFA was used for the first sample, in which principal component analysis (PCA) was performed to extract the factors, and varimax rotation (VR) was used to improve the interpretability of the solution. SPSS software (version 23.0, IBM Corporation, Armonk. NY, USA) was used for this process. For the second sample, AMOS software (Version 24.0, IBM Corporation, Armonk, NY, USA) was used for CFA and SEM. In this part, we performed SEM following the two-step approach recommended by Anderson and Gerbing [40]. First, CFA was carried out for each factor to test whether these factors had a significant factor loading index and to analyze the reliability and validity of the questionnaire. Second, based on the hypothesized path model, the SEM parameters were estimated by the maximum likelihood method. In addition, the model was assessed by the following model fit indexes, with the values in parentheses indicating the cutoffs for acceptable fit [41,42]: (1) the chi-square value ($\chi^{2}$); (2) the chi-square degrees of freedom ($\chi^{2}$/DF < 5); (3) the root mean square error of approximation (RMSEA ≤ 0.08); (4) the comparative fit index (CFI ≥ 0.90); (5) the Tucker–Lewis index (TLI ≥ 0.90); and (6) the incremental fit index (IFI ≥ 0.90). All statistical tests were two-sided with the level of significance set at 0.05. Finally, the bootstrap method was used to test the potential mediator effects, and we calculated the total, direct, and indirect effects.

### 3.4. Exploratory Factor Analysis and Research Hypotheses

The EFA results showed that the Kaiser–Meyer–Olkin (KMO) value was 0.966, which was higher than 0.6, indicating the appropriateness of conducting EFA [43]. Further, the result of Bartlett’s test was significant ($\chi^{2}\;=\;$33,912.366, *p* < 0.001), indicating that the relationship among the items was strong and that the data were suitable for EFA [44].

Seven factors with eigenvalues greater than one were extracted from the first EFA, and their cumulative contribution rate reached 64.874%. However, six items were gradually deleted because the factor loadings were less than 0.45 or higher than 0.4 on two or more factors simultaneously [45]: social position of medical personnel, government policy support, number of patients, skill of existing personnel, economy of the city where the PMHC is located, and organizational culture. Finally, seven factors were extracted from the remaining 46 items, and their cumulative contribution rate reached 66.896%. Detailed results of the final EFA are shown in the attachment.

#### 3.4.1. Sense of Gain (SG)

There were 3 items in this factor: professional pride, fulfill personal value and job-related well-being. First, the medical profession has the vital responsibility of providing medical and health services and safeguarding people’s health [46], so the sense of professional pride is high and is closely related to the work quality, job satisfaction, and intention to leave of medical personnel [47,48,49]. Second, in China, the upfront investment of medical personnel, in terms of education and training time, is often longer than that of workers in other industries, and they need considerable knowledge, training, and practice to be competent or obtain professional titles [50]. It is thus particularly important for medical personnel to demonstrate their personal value in their work. Finally, job-related well-being is the emotional response of employees at work. Negative emotions can lead to stress, depression, and anxiety, while positive emotions can help people thrive in the face of difficulties [51]. Moreover, job-related well-being has been identified as a key area for attracting and retaining employees [52], and it also played a decisive role in employment willingness [53], so we proposed that the manifestation of Sense of Gain determined whether medical personnel were amenable to seeking employment at a PMHCI. Therefore, Sense of Gain was used as a dependent variable to explore the relationship between various factors and the willingness of medical personnel seeking employment at PMHCIs.

#### 3.4.2. Remuneration and Development (RD)

This factor included 11 items, including wage, working bonus, social insurance and accumulation fund, position and professional title promotion, etc. These items reflect the concerns about the Remuneration and Development that can be obtained by seeking employment at PMHCIs. Wages and individual development have always been common issues of concern to career groups. For medical personnel, the investment of time and money and the difficulty of obtaining a professional title [50] make them sensitive to remuneration and development issues. In addition, previous studies [26,54,55] have reported that remuneration and development are also important factors that affect the work satisfaction and enthusiasm of community health workers. Therefore, we recommended that the sense of gain of medical personnel seeking employment at PMHCIs can be increased through appropriate remuneration and development opportunities.

#### 3.4.3. Internal Organization Development (IOD)

This factor included 12 items, including department setting, software and hardware facilities, human resource allocation, organizational management system, culture and working environment, etc. Several studies [56,57,58] have documented that an organization’s internal development is fundamental to the turnover tendency, happiness, job satisfaction, and burnout of medical personnel. Besides, this internal development reflects the diagnosis and treatment capabilities of the institution and affects patients’ identification with PMHCIs and willingness to seek treatment [59]. We proposed that, on the one hand, Internal Organization Development is related to medical personnel’s work arrangements, work pressure, and workload and to the Remuneration and Development, which affects the Sense of Gain of medical personnel seeking employment at PMHCIs as well as patients’ willingness to seek medical treatment in PMHCIs.

#### 3.4.4. Condition of the City Where the PMHCI Is Located (CCPL)

There were 7 items in this factor, including the city’s development, environment, transportation, culture and customs, economy, and distance from home, and the reputation of PMHCI where the PMHCI is located. These items reflect the fact that medical personnel not only pay attention to an organization’s internal development but are also concerned about the external environment of the PMHCI. Besides, medical personnel’s wages, individual development, family, and doctor–patient relations are affected by the condition of the city where the PMHCI is located, including economy, culture and customs, which certainly affect their sense of gain as well.

#### 3.4.5. Job Responsibilities (JR)

This factor included five items: work intensity, stress, hours, workload, and post of duty. These items reflect medical personnel’s concern about their specific work content and job responsibilities. Several studies [60,61] have documented that job characteristics are important predictors of job-related happiness. In addition, the arrangement of the work is determined by an organization’s internal development. Therefore, we proposed that Internal Organization Development affects Job Responsibilities and that Job Responsibilities affects the Sense of Gain.

#### 3.4.6. Family Support (FS)

This factor included 4 variables, spouse, children, parents, and house, which represent the family factors considered when medical personnel seek employment at a PMHCI. Studies [62] have pointed out that family support can buffer employees’ job stress and prevent negative work-related outcomes such as job burnout, and family members have also been shown to provide both instrumental and affective support, which positively affect employee’s work life [63]. Medical personnel are no exception; a meta-analysis [64] showed that the conflict between work and family has a strong impact on the high turnover rate of medical personnel and reducing this conflict can improve their happiness. In addition, communication research [65] has shown that the family as a socialization agent conveys both extrinsic and intrinsic work values for developing a professional identity. Therefore, we proposed that Family Support would have a positive impact on the professional identity and work enthusiasm of medical personnel and would inevitably be affected by the Condition of the City Where the PMHCI Is Located via the economy, culture, etc.

#### 3.4.7. Patient Factor (PF)

This factor included local patients’ trust and respect in physicians, the doctor–patient relationship, and the moral character of the patients. Globally, the concept of medical service has changed from being doctor-centric to patient-centric, which reduces physician dominance, advocates greater patient control, and encourages more mutual participation [66]. However, the lack of coordination and conflict between doctors and patients has aroused widespread concern in society and academia, and it is becoming a serious dilemma facing the medical industry and even society as a whole [67]. As such, it is having a substantial impact on the working conditions and psychological pressure experienced by medical personnel [68,69], and it is likely to lead to work fatigue [70]. Studies [71,72] have pointed out that the doctor–patient relationship is also an important factor affecting the resignation or career choices of medical personnel. Therefore, we proposed that the character, trust in PMHCIs, and respect for medical personnel could reduce work pressure and increase enthusiasm and job-related well-being. In addition, the Condition of the City Where the PMHC Is Located factor, as represented by the city’s economy, culture, etc., can also affect patients’ attitude toward PMHCIs.

Based on the above discussion, the following research hypotheses were proposed, and the hypothetical structural model is shown in Figure 1:

**Hypothesis** **1** **(H1).**
*Internal Organization Development has a positive effect on Sense of Gain.*


**Hypothesis** **2** **(H2).**
*Condition of the City Where the PMHCI Is Located has a positive effect on Sense of Gain.*


**Hypothesis** **3** **(H3).**
*Remuneration and Development has a positive effect on Sense of Gain.*


**Hypothesis** **4** **(H4).**
*Family Support has a positive effect on Sense of Gain.*


**Hypothesis** **5** **(H5).**
*Patient Factor has a positive effect on Sense of Gain.*


**Hypothesis** **6** **(H6).**
*Job Responsibilities has a positive effect on Sense of Gain.*


**Hypothesis** **7** **(H7).**
*Internal Organization Development has a positive effect on the Patient Factor.*


**Hypothesis** **8** **(H8).**
*Condition of the City Where the PMHCI Is Located has a positive effect on the Patient Factor.*


**Hypothesis** **9** **(H9).**
*Internal Organization*
*Development has a positive effect on Remuneration and Development.*


**Hypothesis** **10** **(H10).**
*Condition of the City Where the PMHCI Is Located has a positive effect on Remuneration and Development.*


**Hypothesis** **11** **(H11).**
*Internal Organization Development has a positive effect on Job Responsibilities.*


**Hypothesis** **12** **(H12).**
*Condition of the City Where the PMHCI Is Located has a positive effect on Family Support.*


### 3.5. Confirmatory Factor Analysis

CFA was used to test the reliability and validity of our measurement instrument. In the process, nine variables were removed because the standardized factor loadings were lower than 0.7 [73]. These variables were organizational management system, working environment, reputation of the PMHCI, availability of drugs, degree of emphasis on clinical, institution size, level of knowledge, distance from home, and post of duty.

Finally, as shown in Table 1, all the Cronbach’s alpha and composite reliability (CR) values were above 0.8, indicating acceptable reliability for all constructs. Further, the average variance extracted (AVE) value of each construct was above 0.5 [74], and the standardized factor loading of each item was above 0.7 [41], indicating good convergent validity. In addition, as shown in Table 2, the discriminant validity was verified because the square roots of the AVEs of each construct were higher than their correlation [75]. Therefore, we concluded that the remaining items had sufficiently good reliability and validity to test the structural model of our proposed hypotheses.

Since the data for all constructs were collected using the same measurement instrument, we tested the possibility of common method bias. First, the values of the correlation coefficients in Table 2 were all lower than 0.9, indicating that there were no pairs with very strong correlations [41]. Second, the Harman single-factor test was conducted by PCA, and the results showed that the first extracted factor in the unrotated solution accounted for 41.02% of the variance, which was less than 50% [76]. Finally, controlling for the effects of an unmeasured latent methods factor, common method bias was tested for. As shown in Table 3, after adding the common method factor, the variance of the fit index was very small, and even the RMSEA value decreased. Therefore, common method bias did not seem to affect the result.

### 3.6. Structural Model Analysis

Table 4 presents the fit of the structural equation model. As can be observed, the hypothesized model fits the data well. The *t*-values of each path were computed to test the hypothesized relationships in our research model in AMOS, and the results are shown in Figure 2. In the hypothesized model, Internal Organization Development (*β* = 0.154; *p* < 0.001), the Patient Factor (*β* = 0.547; *p* < 0.001), Remuneration and Development (*β* = 0.129; *p =* 0.004), and Family Support (*β* = 0.081; *p* = 0.018) had a significantly positive effect on the Sense of Gain of medical personnel seeking employment at PMHCIs. Therefore, H1, H3, H4, and H5 were supported, whereas the hypotheses regarding Job Responsibilities (*β* = 0.055; *p* = 0.053) and Condition of the City Where the PMHCI Is Located (*β* = 0.022; *p* = 0.652) were not supported. In addition, both Internal Organization Development (*β* = 0.377; *p <* 0.001, *β* = 0.344; *p* < 0.001) and Condition of the City Where the PMHCI Is Located (*β* = 0.460; *p* < 0.001, *β* = 0.497; *p* < 0.001) had significantly positive effects on Patient Factor and Remuneration and Development, so H7, H8, H9, and H10 were all supported. Finally, Internal Organization Development (*β* = 0.283; *p* < 0.001) had a significantly positive effect on Job Responsibilities, and Condition of the City Where the PMHCI Is Located (*β* = 0.523; *p* < 0.001) had a significantly positive effect on Family Support, indicating that H11 and H12 were supported. The results provide a useful theoretical perspective for taking corresponding measures to guide medical personnel toward work in PMHCIs.

In addition, to test the mediating role of the Patient Factor, Family Support, Job Responsibilities, and Remuneration and Development, we applied the bootstrapping technique in AMOS. A 95% confidence interval of the indirect effects was obtained with 5000 bootstrap resamples. As shown in Table 5, the indirect effects of Internal Organization Development and Condition of the City Where the PMHCI Is Located on Sense of Gain were significant. The Patient Factor, Family Support, and Remuneration and Development significantly mediated the relationship between the internal and external environment of the institution and Sense of Gain, whereas the mediating effect of Job Responsibilities was not significant.

## 4. Discussion

In this paper, we investigated the factors that influence medical personnel to seek employment in PMCHIs based on implicit theory and a lexical approach. Through a pre-investigation and investigation, 7 factors were obtained—namely, Sense of Gain, Remuneration and Development, Internal Organization Development, Condition of the City Where the PMHCI Is Located, Job Responsibilities, Family Support, and the Patient Factor. In addition, SEM was applied to explore the interrelationships and path of each factor. Our results are sound and robust for the following reasons. First, a pre-investigation was used to collect the factors, and an investigation was used to propose and verify the theoretical hypotheses, so the research design was highly scientific. Second, the soundness of the data collection process was guaranteed by several measures, such as screening questions, IP confirmation, answering time, and empty item review. Therefore, the authors believe that this paper provides a theoretical reference allowing the government and PMHCIs to take adopt measures to strengthen the development of health human resources in PMHCIs.

The results show that the Remuneration and Development factor had the most items, indicating that medical personnel paid the most attention to it and that it positively affected the willingness of medical personnel to turn to PMHCIs for employment. This factor not only represents a common requirement of job seekers but also relates to the specific remuneration and development status in PMHCIs. In other words, the low remuneration and poor individual development opportunities of medical personnel have increased job burnout and the turnover rate and have resulted in low satisfaction, serious attrition among medical personnel, and difficulty recruiting personnel to PMHCIs. Therefore, remuneration and development have become an important focus for medical personnel seeking employment in PMHCIs. More importantly, we found that the Internal Organization Development and Condition of the City Where the PMHCI Is Located factors had a significantly positive effect on Remuneration and Development, indicating that the optimization of the internal and external conditions of the institution with respect to department construction, facilities, learning atmosphere, and scientific research and the development and environment of the city where a PMHCI is located can increase perceptions of remuneration and development among medical personnel in PMHCIs. Therefore, when guiding medical personnel to work in PMHCIs, it is not sufficient to consider remuneration and development separately; this factor should be analyzed in conjunction with the development of the institution and of the city. The absence of any one subfactor could reduce the motivation of medical personnel to turn to PMHCIs for employment, and only when the three factors reach a certain level can primary medical institutions successfully recruit and retain medical personnel. In short, complete and excellent development of institutions and cities as well as remuneration and development opportunities of corresponding quality can effectively encourage medical personnel to work in PMHCIs.

The Patient Factor was an unexpected factor that significantly affected the Sense of Gain of medical personnel seeking employment in PMHCIs, and it provided a new perspective for guiding medical personnel to work in such institutions. By analyzing the items contained in the Patient Factor, we found that patients’ trust and respect toward medical personnel were an important embodiment of professional pride and value, and the patient–doctor relationship and moral character of the patients were also important aspects of job-related well-being, so the Patient Factor was closely related to the Sense of Gain factor. In addition, the subfactors of Internal Organization Development, such as software and hardware facilities and department setting, positively affect patients’ trust and respect. In addition, the culture, folk customs, and development of the city where the institution is located also positively affect patients’ moral character, the doctor–patient relationship, and their trust in PMHCIs. Therefore, we propose that the improvement of PMHCIs development and the continuous improvement of patients’ awareness of the role of PMHCIs in the hierarchical diagnosis and treatment system can increase patients’ trust in and satisfaction with PMHCIs, thereby increasing the willingness of medical personnel to turn to PMHCIs for employment.

Family Support had a positive and significant impact on Sense of Gain, indicating that if factors such as family members and housing for medical personnel were fully considered, the willingness of medical personnel to seek employment at PMHCIs would increase. Moreover, Condition of the City Where the PMHCI Is Located positively and significantly affected Family Support, suggesting that the living and working conditions of medical personnel’s family members need to be considered when guiding medical personnel to turn to PMHCIs for employment.

Job Responsibilities was an important factor influencing the willingness of medical personnel to seek employment at PMHCIs, but the results showed that the correlation between Job Responsibilities and Sense of Gain was not significant. The main reason is likely that most of the participants who were facing job selection or undergoing pre-job selection training were in large hospitals with scientific research and teaching qualifications, and the gap between a large hospital and PMHCIs in terms of work pressure, workload, and work difficulty is large. Therefore, the medical personnel had less awareness of the specific job responsibilities at PMHCIs. However, against the backdrop of integrating public health services and clinical services, the medical personnel in PMHCIs have taken on expanded roles [77,78], so they are under considerable pressure. In addition, it has been shown that workload, work intensity, and work pressure are important factors that affect the satisfaction and willingness to quit of medical personnel in PMHCIs [56,70,79]. Therefore, this study proposes that although the participants had not yet clarified the relationship between their job responsibilities and sense of gain, Job Responsibilities is still an important factor affecting the work of medical personnel in PMHCIs.

The relationship between Internal Organization Development and Sense of Gain was positively significant, indicating that departments with a scientific setting, software and hardware facilities, and a higher scientific research level improve the work experience of medical personnel and thereby improve the willingness of medical personnel to seek employment at a PMHC. In addition, Internal Organization Development indirectly affected Sense of Gain through the Patient Factor and Remuneration and Development. Thus, the scientific and complete development of the internal organization of PMHCIs not only affects the willingness of medical personnel directly but also improves patients’ trust in and respect for PMHCIs, increases their remuneration, and creates individual development opportunities, all of which improve the willingness of medical personnel to seek employment at a PMHCI.

Regarding the insignificant effect of Condition of the City Where the PMHCI Is Located on Sense of Gain, it may be that compared to the weight given to the external environment, medical personnel pay more attention to proximal factors such as individuals, family, and internal development. However, through the analysis of the mediation effect, the influence of the condition of the city on personnel’s sense of gain cannot be ignored; that is, it can affect Sense of Gain through the Patient Factor and Remuneration and Development. Therefore, we conclude that Condition of the City Where the PMHC Is Located has an indirect and vitally important impact on the willingness of medical personnel to seek employment in PMHCIs.

### Strengths and Limitations of This Study

The major strengths of this study include the following aspects. First of all, this study took as its research object candidates for the medical licensing examination who were facing graduation or career choice and were thus a suitable group to be guided to work in PMHCIs. Previous studies had not paid attention to this population, so the study provides a new perspective for improving the human resources development of PMHCIs. Second, the analysis from the perspective of the guided object solves the defect that the original measures were only formulated from the perspective of the government and PMHCIs and lacked understanding of the actual needs of medical personnel. Third, this study analyzed factors such as the internal and external environment of the institution, family, patients, remuneration, and individual development on the willingness of medical personnel to seek employment at PMHCIs, and the factors involved were comprehensive and representative. However, this study also has three limitations. First, because of the characteristics of the candidates, they did not represent all categories of medical personnel. Second, attitudes vary from region to region, so further analysis based on local conditions is needed. Third, this study was a cross-sectional analysis, making it impossible to determine changes in the attitudes of medical personnel over time.

## 5. Conclusions

This study used implicit theory and a lexical approach to explore the factors that affect candidates who were facing career choices to seek employment at PMHCIs, and a structural equation mode was established to analyze the relationship between the factors.

We conclude that the improvement of Family Support, Remuneration and Development, and Patient Factors will increase the willingness of medical personnel to turn to PMHCIs for employment. In addition, the Internal Organization Development factor has both direct and indirect effects on this willingness through the Patient Factor and Remuneration and Development, whereas Condition of the City Where the PMHCI Is Located has indirect effects on willingness through the Patient Factors, Remuneration and Development, and Family Support. This research provides theoretical support for improving the development of human resources, guiding medical personnel to work in PMHCIs, and promoting the use of primary care services.

## Figures and Tables

**Figure 1 ijerph-18-02560-f001:**
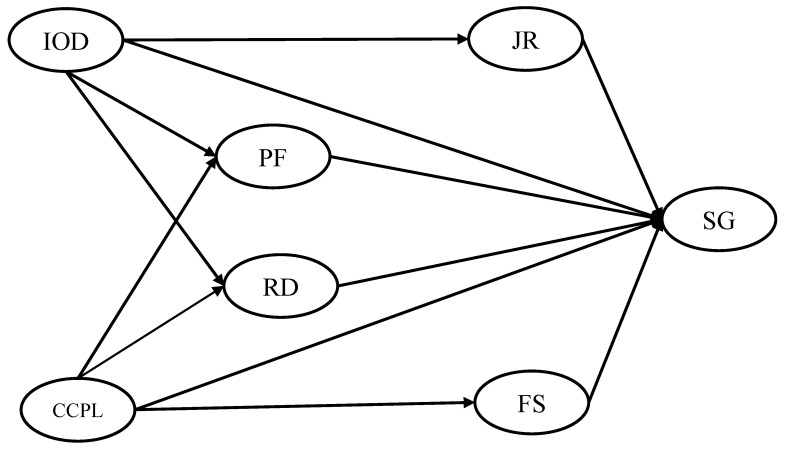
The hypothetical structural model of the influencing factors. SG, Sense of Gain; RD, Remuneration and Development; IOD, Internal Organization Development; CCPL, Condition of the City Where the PMHCI Is Located; JR, Job Responsibilities; FS, Family Support; PF, Patient Factor.

**Figure 2 ijerph-18-02560-f002:**
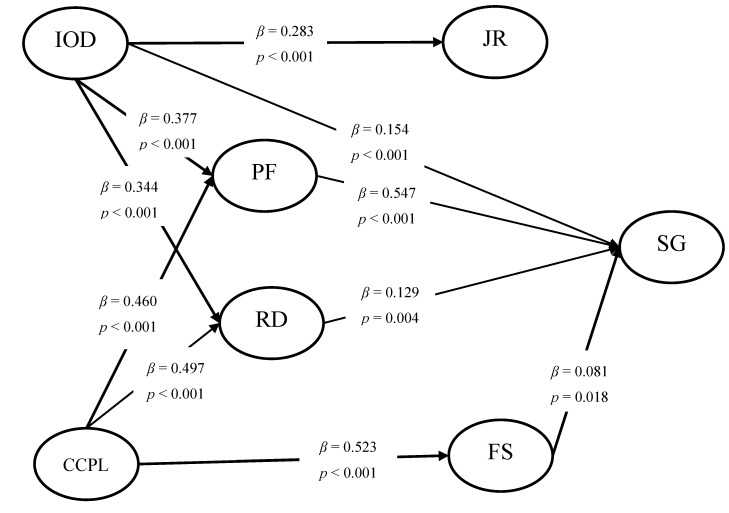
The structural model of the influencing factors.

**Table 1 ijerph-18-02560-t001:** Reliability and validity test.

Factors	Items	Standardized Factor Loading	CR ^a^	AVE ^b^	Cronbach’s Alpha
Condition of the City Where the PMHCI Is Located CCPL	Development	0.866	0.897	0.637	0.861
	Environment	0.848			
	Culture and Customs	0.762			
	Transportation	0.758			
	Location	0.748			
Remuneration and Development RD	Working Bonus	0.867	0.951	0.638	0.951
	Medical Insurance	0.853			
	Working Subsidy	0.846			
	Performance Assessment	0.807			
	Professional Title Promotion	0.803			
	Social Insurance and Accumulation	0.792			
	Wage	0.787			
	Holidays Arrangements	0.779			
	Authorized Strength	0.754			
	Position Promotion	0.749			
	Individual Development	0.737			
Internal Organization Development IOD	Software and Hardware Facilities	0.835	0.897	0.592	0.895
	Specialist Construction	0.801			
	Learning Atmosphere	0.788			
	Human resource allocation	0.752			
	Teaching and scientific research	0.727			
	Department Setting	0.706			
Job Responsibilities JR	Working Intensity	0.921	0.909	0.715	0.907
	Working Stress	0.889			
	Working Hours	0.782			
	Workload	0.780			
Family Support FS	House	0.858	0.883	0.654	0.882
	Parents	0.814			
	Children	0.784			
	Spouse	0.775			
Patient Factor PF	Respect in Physician	0.832	0.866	0.617	0.862
	Patient–Doctor Relationship	0.806			
	Trust in Physician	0.760			
	Moral Character of Patients	0.741			
Sense of Gain SG	Fulfilling Personal Value	0.849	0.859	0.670	0.858
	Professional Pride	0.833			
	Job-Related Well-being	0.772			

^a^ Composite Reliability. ^b^ Average Variance Extracted.

**Table 2 ijerph-18-02560-t002:** Discriminant validity of constructs.

	CCPL ^a^	RD ^b^	IOD ^c^	JR ^d^	FS ^e^	SG ^f^	PF ^g^
CCPL	0.798						
RD	0.689	0.799					
IOD	0.610	0.637	0.769				
JR	0.305	0.232	0.262	0.845			
FS	0.498	0.459	0.329	0.275	0.808		
SG	0.597	0.625	0.611	0.209	0.448	0.819	
PF	0.669	0.694	0.645	0.319	0.512	0.770	0.786

^a^ Condition of the City Where the PMHCI Is Located. ^b^ Remuneration and Development. ^c^ Internal Organization Development. ^d^ Job Responsibilities. ^e^ Family Support. ^f^ Sense of Gain. ^g^ Patient Factor.

**Table 3 ijerph-18-02560-t003:** Common method bias test.

	$\chi^{2}$	df	$\chi^{2}$/df	RMSEA ^a^	CFI ^b^	TLI ^c^	IFI ^d^
Without common method factor	2535.443	608	4.170	0.060	0.921	0.913	0.921
With common method factor	2507.799	661	4.100	0.059	0.922	0.915	0.922

^a^ The Root Mean Square Error of Approximation. ^b^ The Comparative Fit Index. ^c^ The Tucker–Lewis Index ^d^ The Incremental Fit Index.

**Table 4 ijerph-18-02560-t004:** The fit of the structural equation model.

Fit Indices	$\chi^{2}$	df	$\chi^{2}/\mathrm{df}$	RMSEA ^a^	CFI ^b^	TLI ^c^	IFI ^d^
Value in the models	1988.127	611	3.254	0.050	0.943	0.938	0.943

^a^ The Root Mean Square Error of Approximation. ^b^ The Comparative Fit Index. ^c^ The Tucker–Lewis Index ^d^ The Incremental Fit Index.

**Table 5 ijerph-18-02560-t005:** Analysis of mediating effect.

Path	Effect Sizes	Boot SE	Z	*p*	Bias-Corrected 95%CI		Percentile 95%CI	
					Lower	Upper	Lower	Upper
Int1 ^a^	0.252	0.046	5.478	<0.001	0.169	0.351	0.167	0.347
Int2 ^b^	0.042	0.021	2.000	0.046	0.004	0.087	0.001	0.084
Int3 ^c^	0.064	0.028	2.286	0.011	0.011	0.123	0.009	0.120
Int4 ^d^	0.206	0.037	5.568	<0.001	0.142	0.289	0.140	0.286
Int5 ^e^	−0.016	0.011	−1.455	0.147	−0.038	0.004	−0.038	0.004
Int6 ^f^	0.044	0.020	2.200	0.028	0.008	0.089	0.006	0.086
Total Indirect Effect	0.593	0.069	8.594	<0.001	0.462	0.732	0.457	0.730

^a^: Condition of the City where the PMHCI Is Located → Patient Factor→ Sense of Gain; ^b^: Condition of the City Where the PMHCI Is Located → Family Support → Sense of Gain; ^c^: Condition of the City Where the PMHCI Is Located → Remuneration and Development → Sense of Gain; ^d^: Condition of the City Where the PMHCI Is Located → Patient Factor → Sense of Gain; ^e^: Condition of the City Where the PMHCI Is Located → Job Responsibilities → Sense of Gain; ^f^: Condition of the City Where the PMHCI Is Located→ Remuneration and Development→ Sense of Gain.

## Data Availability

The data presented in this study are available on request from the corresponding author. The data are not publicly available due to personal privacy.

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
