# Peer review of "A New Perspective for Improving the Human Resource Development of Primary Medical and Health Care Institutions: A Structural Equation Model Study"

_ijerph, 2021, doi:10.3390/ijerph18052560_

Round 1

Reviewer 1 Report

This study of factors that influence medical personnel to seek employment in PCHMIs was through and organized. Minor changes include:

line 78, change cultural to culture

Line 84 change research object to research subjects

Line 94: more explanation should be given to explain lexical, which is not usually a construct used in such studies but is an interesting idea

Line 183: You state the at 1778 completed and valid responses were received. What made a response complete and valid and how to complete and valid differ from one another? How many paper questionnaires were distributed and what was the percent response rate?

Author Response

Dear Reviewer,

We appreciate for your constructive comments on our paper. We would like to submit the revised manuscript entitled “A new perspective for improving the human resource development of primary medical and health care institutions: A structural equation model study” (manuscript ID: ijerph-1131178). We have carefully revised the manuscript text based on your comments, and the "Track Changes" function was used to mark the changes.  If there are other questions about the paper, I hope you will not hesitate to raise them to help us improve the quality of my paper and publish in “International Journal of Environmental Research and Public Health”. The following is the point-by-point responses to your comments.

Thank you again for your generosity and best regards.

Sincerely,

Xihe Yu

Point 1: line 78, change cultural to culture

Response 1: We have changed "cultural" to "culture". (p. 2, lines 80).

Point 2: Line 84 change research object to research subjects

Response 2: We have changed "research object" to "research subjects". (p. 2, lines 86).

Point 3: Line 94: more explanation should be given to explain lexical, which is not usually a construct used in such studies but is an interesting idea

Response 3: We have supplemented the explanation of the lexical approach, especially explaining the application in understanding the people's opinions on the organization entity, which echoes the application of the lexical approach to understand the candidates’ attitudes and views towards PMHCIs in this paper. (p. 3, lines 147-155).

Point 4: Line 183: You state the at 1778 completed and valid responses were received. What made a response complete and valid and how to complete and valid differ from one another? How many paper questionnaires were distributed and what was the percent response rate?

Response 4: In the investigation, we asked the administrator of the Changchun Medical Examination Center to coordinate and encourage candidates to participate in our investigation without affecting the candidates' examination, so the participation of the candidates was very high. In the analysis, in order to ensure the integrity of the questionnaire without losing any of the candidates’ answers, we deleted the questionnaires that are incomplete or have the same answers of the whole questionnaire, so as to ensure the scientific analysis results. Finally, a total of 1,883 questionnaires were distributed, all questionnaires were returned, and 105 questionnaires were incomplete or had the same answers for all items, which have been deleted from the analysis (the completion rate was 94.42%). Based on your point, we had added the results of the questionnaire distribution and response in the paper. (p. 4, lines 238-240).

Reviewer 2 Report

The manuscript is well written and the result is clear.  However, there are several shortcomings. The reviewer’s concerns are as follows;

  1. L146: What is “IP confirmation”?
  2. L326-344: The readers would understand the hypothesis more clearly when using the original SEM model. Only final model (Fig.1) is described in the text.
  3. Table 1: Why is the “CCPL (Development)” is underlined?
  4. Table 1: According to the factor analysis, IOD factor include 12 items (Line 268).  However, factor loadings of only 6 items are written in Table 1. Please explain.
  5. Fig 1: Usually, non-significant relationships, between “JR and SG” and “CCPL and SG” are deleted.
  6. Table 5: Why is the “Int 1” is underlined? The lower values of 95% CI are listed, however upper values are not. Is it correct?
  7. L525-542: Conclusion is too long.
  8. The style of the reference is not unified. For example, there are abbreviated and full journal names. When is the access date (Ref. No12 and 14)? The journal title of No24, “Fam Med Community He” is correct? Since there are relatively many references, the reviewer recommends to reduce the number of references.

Author Response

Dear Reviewer,

We appreciate for your constructive comments on our paper. We would like to submit the revised manuscript entitled “A new perspective for improving the human resource development of primary medical and health care institutions: A structural equation model study” (manuscript ID: ijerph-1131178). We have carefully revised the manuscript text based on your comments, and the "Track Changes" function was used to mark the changes.  If there are other questions about the paper, I hope you will not hesitate to raise them to help us improve the quality of my paper and publish in “International Journal of Environmental Research and Public Health”. The following is the point-by-point responses to your comments.

Thank you again for your generosity and best regards.

Sincerely,

Xihe Yu

Point 1: L146: What is “IP confirmation”?

Response 1: In the pre-investigation, a data collection service from a leading Chinese online survey site was adopted to administer the pre-investigation, which can provide the IP address of the respondent. "IP confirmation" means preventing one respondent from filling out the survey multiple times and people from other regions from filling out the survey by using the same computer and geographic restrictions. This is a preliminary review of the data collected.

Point 2: L326-344: The readers would understand the hypothesis more clearly when using the original SEM model. Only final model (Fig.1) is described in the text.

Response 2: We have added the original SEM model in the hypothesis. (p. 8, lines 438).

Point 3: Table 1: Why is the “CCPL (Development)” is underlined?

Response 3: The lines in Table 1 are just trying to separate each factor to make it easier for readers to read, without other meanings. To avoid ambiguity, we had optimized the table. (p. 8, lines 455).

Point 4: Table 1: According to the factor analysis, IOD factor include 12 items (Line 268).  However, factor loadings of only 6 items are written in Table 1. Please explain.

Response 4: According to 3.3 Data Analysis, the exploratory factor analysis (EFA) was used to extract common factors from the items and obtain concise and representative factors, and the results of the final EFA are shown in the attachment due to the large size of the table. The confirmatory factor analysis (CFA) was adopted to test the reliability and validity of the dimensions, which is the prerequisite for establishing structural equation models. In CFA, nine variables were removed because the standardized factor loadings were lower than 0.7, and among these nine variables, the Organizational Management System, Working Environment, Availability of Drugs, Degree of Emphasis on Clinical, Institution Size, and Level of Knowledge belong to IOD in the results of EFA. Therefore, Table 2 shows the finally results of CFA, which had deleted the nine variables.

Point 5: Fig 1: Usually, non-significant relationships, between “JR and SG” and “CCPL and SG” are deleted.

Response 5: We have deleted the non-significant relationships, between “JR and SG” and “CCPL and SG” in Figure 2. (p. 11, lines 496).

Point 6: Table 5: Why is the “Int 1” is underlined? The lower values of 95% CI are listed, however upper values are not. Is it correct?

Response 6: We appreciate for your reminder. This is an error in table making, and we have revised it. (p. 12, lines 511).

Point 7: L525-542: Conclusion is too long.

Response 7: Based on your opinion, we appropriately reduced the Conclusion. (p. 14, lines 635-647).

Point 8: The style of the reference is not unified. For example, there are abbreviated and full journal names. When is the access date (Ref. No12 and 14)? The journal title of No24, “Fam Med Community He” is correct? Since there are relatively many references, the reviewer recommends to reduce the number of references.

Response 8: We have added the access date (Ref. No12 and 14). According to the instructions of “International Journal of Environmental Research and Public Health” EndNote had been used to cite the reference, and based on your comments, we have carefully checked and proofread the name of all journals. In addition, we appropriately reduced some references.

Round 2

Reviewer 2 Report

The  manuscript was revised correctly.